# Substrate Type and Concentration Differently Affect Colon Cancer Cells Ultrastructural Morphology, EMT Markers, and Matrix Degrading Enzymes

**DOI:** 10.3390/biom12121786

**Published:** 2022-11-30

**Authors:** Marco Franchi, Konstantinos-Athanasios Karamanos, Concettina Cappadone, Natalia Calonghi, Nicola Greco, Leonardo Franchi, Maurizio Onisto, Valentina Masola

**Affiliations:** 1Department for Life Quality Studies, University of Bologna, 47921 Rimini, Italy; 2Department of Pharmacy and Industrial Pharmacy, University of Bologna, 40100 Bologna, Italy; 3Department of Pharmacy and Biotechnologies, University of Bologna, 40126 Bologna, Italy; 4Department of Biomedical Sciences, University of Padova, 35131 Padova, Italy; 5Department of Medicine, University of Bologna, 40126 Bologna, Italy

**Keywords:** colon cancer, doxorubicin, matrigel, type I collagen, matrix degrading enzymes, 3D cell cultures, epithelial-to-mesenchymal transition (EMT), scanning electron microscopy (SEM)

## Abstract

Aim of the study was to understand the behavior of colon cancer LoVo-R cells (doxorubicin-resistant) vs. LoVo-S (doxorubicin sensitive) in the initial steps of extracellular matrix (ECM) invasion. We investigated how the matrix substrates Matrigel and type I collagen-mimicking the basement membrane (BM) and the normal or desmoplastic lamina propria, respectively-could affect the expression of epithelial-to-mesenchymal transition (EMT) markers, matrix-degrading enzymes, and phenotypes. Gene expression with RT-qPCR, E-cadherin protein expression using Western blot, and phenotypes using scanning electron microscopy (SEM) were analyzed. The type and different concentrations of matrix substrates differently affected colon cancer cells. In LoVo-S cells, the higher concentrated collagen, mimicking the desmoplastic lamina propria, strongly induced EMT, as also confirmed by the expression of Snail, metalloproteases (MMPs)-2, -9, -14 and heparanase (HPSE), as well as mesenchymal phenotypes. Stimulation in E-cadherin expression in LoVo-S groups suggests that these cells develop a hybrid EMT phenotype. Differently, LoVo-R cells did not increase their aggressiveness: no changes in EMT markers, matrix effectors, and phenotypes were evident. The low influence of ECM components in LoVo-R cells might be related to their intrinsic aggressiveness related to chemoresistance. These results improve understanding of the critical role of tumor microenvironment in colon cancer cell invasion, driving the development of new therapeutic approaches.

## 1. Introduction

Colorectal cancer (CRC) represents the third most common cancer in men and the second in women worldwide. It is considered the third leading cause of cancer-related death in the world [1]. CRCs are usually adenocarcinomas that develop from benign polyposis and invade the extracellular matrix (ECM) of the large bowel [2]. The main treatment procedures for colon cancer involve chemotherapy and surgery, depending on tumor location, cancer size and stage, as well as the overall characteristics of the patient [1]. Continuous improvements in chemotherapeutic agents increased the overall survival of patients with colon cancer over the past decade [3,4]. The final rate of survival depends on the stage of colon cancer: the five-year survival rates of stage I–IV colon cancer are 96.6%, 88.7%, 69.9%, and 34.3%, respectively. Anyway, adjuvant chemotherapy after radical surgery seems to improve the survival rate of stage II treated patients (90.4%) vs. stage II untreated ones (82.4%) [5]. However, during chemotherapy, colon cancer patients can develop drug resistance which can lead to the failure of treatment [6,7]. For instance, more than 50% of metastatic colorectal cancer patients develop a resistance to 5-fluorouracil, one of the most effective and most commonly selected drugs to treat colorectal cancer [8]. When drug resistance involves more than one drug and includes various chemotherapeutics or also antibiotics, it is defined as multidrug resistance [5,6,7,8,9,10,11,12].

Drug resistance and failure of chemotherapy developed during the drug treatment are related to most cancer mortalities [13]. Even though drug resistance seems to be connected to several factors such as secreted cytokines, higher apoptotic threshold, aerobic glycolysis, hypoxia, and elevated activity of drug efflux transporters, steroid receptors coactivators, and/or receptor tyrosine kinases [14,15], it is worth noticing that cancer cell resistance to chemotherapy is mainly associated with the epithelial-to-mesenchymal transition (EMT), which has been related to increased metastasis [15,16,17,18,19]. In all cancers, as well as in CRC, the EMT process involves changes in both molecular and morphological characteristics and is associated with cell proliferation and drug resistance: epithelial cells dissociate by decreasing their epithelial adhesion molecules such as E-cadherin and increasing the mesenchymal markers such as vimentin [20,21,22,23]. On the other hand, it is well established that the restoration of the epithelial phenotype increases the sensitivity of tumor cells to chemotherapy and improves prognosis [13].

Both molecular and morphological EMT signatures of cancer cells are, in turn, strongly related to the tumor microenvironment (TME) components which, by interplaying with cancer cells, can regulate tumor development, cancer cell invasion, and metastasis [21,24,25,26,27,28,29,30,31,32]. The physical arrangement of the ECM network provides a dynamic bioscaffold for ECM cells and allows the control of many cellular functions and properties, such as proliferation, migration, differentiation, and survival. Similarly, both quantitative and qualitative modifications of ECM composition, as far as changes in intrinsic mechanical properties, are critical for tumorigenesis and cancer progression. Following these considerations, the investigations regarding cancer cell behavior should also include an analysis of ECM, the main part of TME that plays critical regulatory roles in cancer progression [32,33,34,35].

In the CRC microenvironment, the basement membrane (BM) is the first ECM physical barrier, whereas the ECM of lamina propria, mainly containing collagen fibrils, represents a further biological barrier that cells must invade to penetrate the lymphatic or the blood vessels. In CRC peri-tumoral lamina propria, a deposition of type I collagen (desmoplasia) and changes in the collagen fiber array occur to oppose cancer cell invasion. Collagen is the most abundant of all ECM components and assembles in fibers forming a dynamic three-dimensional physical bioscaffold which at first functions like a barrier to prevent cancer cell invasion but, in later stages, can also favor this event [21]. When CRC cells invade the ECM, they activate the surrounding stromal cells, and in particular, the cancer-associated fibroblasts (CAFs), to produce large amounts of type I collagen but also less collagen II, III, IV, and IX in TME; CAFs also secrete the enzyme lysyl oxidase favoring the formation of covalent crosslinks in collagen fibers, making them stiffer [31,36,37]. An alteration of the collagen fiber orientation with a predominance of stiff and aligned fibers vs. randomly arranged ones is representative of a TME favoring cancer invasion [38]. For instance, dense type I collagen fibers and a further orientation of straight collagen fibers perpendicularly to the boundary of tumor mass favor tumor progressive signals, intravasation, metastasis, and poor outcomes in breast cancer [27,39]. Moreover, increased deposition of type I collagen fibers following a particular array seems to promote cancer cells’ survival by physically protecting them against chemotherapy [40,41,42].

Among anticancer drugs which promote chemoresistance, doxorubicin belongs to the antitumor and antibiotic anthracycline family of medications. It works in part by interfering with the function of DNA [10,12]. As a chemotherapeutic agent, it slows or stops the growth of cancer cells by blocking the enzyme topoisomerase II, an enzyme that is essential for cancer cell proliferation. As a chemotherapy medication, it is used to treat cancers such as breast cancer, bladder cancer, Kaposi’s sarcoma, lymphoma, and acute lymphocytic leukemia. The aim of this study was to evaluate the behavior of two colorectal cell types, the sensitive to doxorubicin LoVo-S cells and the multidrug-resistant LoVo-R ones (obtained by prolonged exposure to doxorubicin) in 3D cultures with different ECM substrates [43]. Emphasis was given to screening the cell morphological characteristics in the initial steps of the invasion when cancer cells try to cross the BM or begin to invade the desmoplastic lamina propria. To this end, we decided to use Isopore Membrane Filters with a pore size of 5 µm, which do not allow any cell crossing. In particular, we investigated how the different concentrations of Matrigel and type I collagen, mimicking the BM and normal or desmoplastic lamina propria, respectively, could affect the ultrastructural morphological features and behavior of these two cell types by utilizing the scanning electron microscopy (SEM), the gene expression of EMT markers as well as matrix effectors implicated in cancer propagation and metastasis.

## 2. Materials and Methods

### 2.1. Cell Cultures

Human colorectal adenocarcinoma cell line LoVo-S and its doxorubicin-resistant subline LoVo-R, obtained in vitro through repeated expositions of LoVo cells to 1 µg/mL of the drug, were used. Cells were cultured in RPMI 1640 medium, supplemented with 10% fetal bovine serum (FBS), 2 mM l-glutamine, penicillin (100 U/mL), and streptomycin (100 µg/mL), in humidified air at 37 °C with 5% CO_2_. The resistant phenotype was obtained by continuous treatments of cells with doxorubicin (0.1 µg/mL every 5 passages). Resistance to doxorubicin is verified before each experiment, as previously reported [44,45].

To mimic structural ECM natural barriers cells at 80% of confluences were detached with Trypsin-EDTA solution and (1.5 × 10^5^) were seeded for 24 h on “Isopore Membrane Filters” with a pore size of 5.0 µm (Millipore, Milan, Italy) coated with Matrigel (BD Biosciences, Milan, Italy) or type I Collagen solution (C3867, Sigma-Aldrich, Schnelldorf, Germany). Collagen and Matrigel filter coating was performed by diluting Matrigel and Collagen at the proper concentrations (0.2 or 3.5 mg/mL) in sterile water (pH 6), distributed on the filter, and incubated for 2 h at 37 °C for polymerization. The lower chamber was filled with F-12K 20% FBS.

To evaluate LoVo-S/-R cells at the first steps of ECM invasion, therefore, when they just try to cross the Matrigel or begin to invade the collagen meshwork, we decided to use Isopore Membrane Filters with a pore size of 5.0 µm, which represents a limit to nuclear deformations and do not allow any cell crossing.

### 2.2. RNA Isolation and Real-Time qPCR Analysis

Total RNA was extracted from cells with a Trizol reagent (Invitrogen, Thermo Fisher, Waltham, MA, United States) according to the manufacturer’s instructions [46]. RNA yield and purity were checked using a Nanodrop spectrophotometer (EuroClone, Milan, Italy), and total RNA from each sample was reverse transcribed into cDNA using Moloney Murine Leukemia Virus Reverse Transcriptase (Sigma-Aldrich). Real-time PCR was performed on a StepOne™ Real-Time PCR System (Thermo Fisher, Waltham, MA, USA) using SensiFAST SYBR Hi-Rox (Bioline, LABGENE SCIENTIFIC SAZI, Châtel-Saint-Denis, Switzerland). The comparative Ct method (ΔΔCt) was used to quantify gene expression, and the relative quantification was calculated as 2−ΔΔCt. The presence of non-specific amplification products was excluded by melting curve analysis. Statistical analyses on real-time PCR data were performed using the Relative Expression Software Tool (REST2009, Qiagen, Venlo, The Netherlands) [47]. The forward and reverse primer sequences are shown in Table 1.

### 2.3. E-Cadherin Expression by Western Blot

To evaluate protein expression, LoVo-S and LoVo-R cells were seeded (1.5 × 10^5^ cells/cm^2^) on Millipore filters covered with Matrigel or Collagen at different concentrations (0.2 or 3.5 mg/mL). After 24 h, the medium was removed, cells were washed with PBS and were lysed in RIPA buffer composed of 50 mM Tris–HCl, pH 5.0, 150 mM NaCl, 0.5% Triton X-100 with Complete Protease Inhibitor Mixture (Roche Applied Science, Penzberg, Germany). After quantification, equal amounts of proteins were treated in reducing sample buffer and denatured for 10 min at 100 °C. Protein samples were then resolved in 10% SDS–PAGE and electrotransferred to nitrocellulose membranes. Non-specific binding was blocked for 1 h at room temperature with non-fat milk (5%) in TBST buffer (50 mM Tris–HCl, pH 7.4, 150 mM NaCl, and 0.1% Tween 20). Membranes were exposed to primary antibodies (1:1000) directed against GAPDH (sc-47778 Santa Cruz) and E-CADHERIN (E-CAD) (GTX10443 GeneTex, Irvine, CA, USA) overnight at 4 °C and incubated with a secondary peroxidase-conjugated antibody for 1 h at room temperature. The signal was detected with Luminata™ Forte Western HRP Substrate (Millipore) according to the manufacturer’s instructions, and the signal was acquired with Mini HD9 (UVItec, Cambridge, UK).

### 2.4. Statistical Analysis

Statistical Analysis Reported values are expressed as mean ± standard deviation (SD) of experiments performed in triplicate. Statistically significant differences were evaluated using the analysis of a two-tailed Student’s *t*-test and were considered statistically significant at the level of at least *p* ≤ 0.05. Statistical analysis was performed using Rest2009 software (Qiagen).

### 2.5. Scanning Electron Microscopy

LoVo-S and LoVo-R colon cancer cells were seeded on six isopore membrane filters (Millipore, Milan, Italy) with 5 µm pore size and on similar 24 filters covered by different biological substrates (Matrigel or type I collagen) at different concentrations (0.2 or 3.5 mg/mL) for 3D cultures. After 24 h of culturing, the samples, including Millipore filters and cells, were fixed in a Karnovsky’s solution for 20 min. The samples were rinsed three times with 0.1% cacodylate buffer, dehydrated with increasing concentrations of ethanol, and finally dehydrated with hexamethyldisilazane (Sigma-Aldrich, Inc.) for 15 min. The specimens were mounted on appropriate stubs, coated with a 5 nm palladium gold film (Emitech 550 sputter-coater, Quorum Technologies, Lewes, United Kingdom) to be observed under a scanning electron microscope (SEM) (Philips 515, Eindhoven, The Netherlands) operating in secondary-electron mode.

## 3. Results

### 3.1. Evaluation of EMT Markers and Matrix Degrading Enzymes in CRC Cells Cultured in Different Matrix Substrates

To evaluate how matrix substrates could affect the aggressiveness of the two CRC cell types, LoVo-S and LoVo-R, we first screened the expression of the EMT-related genes E-cadherin, vimentin, and Snail. This was performed by collecting cultures of LoVo-S and LoVo-R cells for 24 h on a Millipore filter (5 μm pore size) covered either by Matrigel or type I collagen at different concentrations, isolation of total RNA and RT-qPCR analysis. Millipore filters with 5 μm pore size limit cell migration and dispersion. This is a useful tool to observe LoVo cancer cells preparing to invade the ECM components used as substrates. As shown in Figure 1A, the epithelial marker E-cadherin exhibited a much higher expression in LoVo-S as compared to LoVo-R cells. Particularly, a 2.5-times higher expression in LoVo-S cells vs. LoVo-R was noted in the lower concentration of Matrigel (0.2 mg/mL) and 3.2-times higher in the highly concentrated type I collagen group (3.5 mg/mL). Differently, the E-cadherin expression in LoVo-R cells was not significantly affected with respect to both type and concentration of the matrix substrate in all groups (Figure 1A). These data were also confirmed at the protein level using Western blot analysis (Figure 1B). The expression of the mesenchymal marker vimentin was significantly higher (*ca* 20 to 100 times) in LoVo-R cells as compared to LoVo-S (Figure 1C). Similarly, also Snail was significantly more expressed in LoVo-R vs. LoVo-S cells (Figure 1D). Conclusively, these data suggest that the type of matrix substrate and its concentration could significantly affect the EMT program of LoVo-S (less aggressive) as compared to LoVo-R cells (highly aggressive).

Moving forward, we evaluated the expressions of metalloproteases (MMPs) to define which substrate could greatly affect the behavior of LoVo cancer cells when they organize to invade the ECM. We observed that when LoVo-S cells prepared themselves to cross and invade Matrigel or collagen at different concentrations, the MMP-2, -9, and -14 were highly expressed (3 to 5 times) in the highly concentrated collagen groups vs. all the other substrates, which showed lower and similar values (Figure 1E–G). HPSE was also highly expressed in highly concentrated collagen groups even though a significant value was also measured in LoVo-S cells cultivated in concentrated Matrigel vs. the low concentrated one. Similarly, MMP-9 was relatively higher (50%) in concentrated Matrigel substrate vs. the lower concentrated one. These data demonstrate that in LoVo-S cells, the MMP-9 and HPSE expressions mainly depend on substrate concentration (Figure 1F,H).

Definitely, in LoVo-S cells the highly concentrated collagen meshwork (3.5 mg/mL), corresponding to a desmoplastic lamina, the ECM substrate induces a substantial increase in the expression of almost all the tested aggressive parameters (Snail, MMP-2, -9, -14 and HPSE) (Figure 1D–H), but also the highest E-cadherin expression (Figure 1A) and a vimentin suppression (Appendix A).

Interestingly, in LoVo-R cells the highly concentrated collagen does not appreciably influence the E-cadherin, vimentin, Snail, and MMP-2, -14 expressions when compared to the other substrates, suggesting that neither the type of substrate nor substrate concentration significantly affects these gene expressions (Figure 1A,C–G). The MMP-9 expression was not the highest in the highly concentrated collagen as it was more expressed in concentrated/thicker Matrigel groups, suggesting that also in LoVo-R cells, like in LoVo-S ones, the MMP-9 expression depends on the substrate concentration (Figure 1F). In addition, only a slight increase in HPSE expression was found in highly concentrated collagen compared to all the other substrates (Figure 1H).

### 3.2. Ultrastructural Morphological Features of CRC Cells Cultured on Millipore/Matrigel Mimicking a Normal and a Thick Basement Membrane

Taking into consideration the significant differences in gene expression of EMT markers and matrix-degrading enzymes showed by the LoVo-S/-R colon cancer cells growing in different matrix substrates at different concentrations, we further evaluated their ultrastructural phenotype using SEM. In order to examine the phenotypes that the cells exhibit when they are moving to cross the ECM components, we used a Millipore filter with a 5 µm pore size, which does not allow cell crossing.

When LoVo-S cells were cultured in a Millipore filter, they appeared grouped, and all of them showed cell-cell contacts. They developed both a partially flattened phenotype (10–20 µm in diameter) rather than a globular-shaped one. About 6% of LoVo-S cells showed fewer cell-cell contacts and exhibited a concave conic/funnel shape, suggesting that they were trying to pass through the pores of the Millipore filter (Figure 2A,B). All cells exhibited sparse microvilli and no microvesicles (Figure 2C). On the other hand, all LoVo-R cells appeared like smooth, regular, and flattened cobblestone-shaped cells (10–20 µm in diameter); but at a tilted observation they appeared like concave leaves when trying to cross the small pores of the Millipore filter by developing filopodia. Even though the cells looked grouped, they did not show tight cell-cell contacts (Figure 2D–F).

Then we evaluated the ultrastructural morphology of LoVo cells which are preparing to invade the BM barrier. To this purpose, both LoVo-S and LoVo-R cells were cultivated on a Millipore filter covered by Matrigel at a standard concentration of 0.2 mg/mL. Both LoVo-S and LoVo-R cells showed only remnants of the Matrigel. A portion of *ca* 65–70% (*p* < 0.05%) of LoVo-S cells exhibited an epithelial-like phenotype, including large and very flattened polygonal cells (10–20 µm) which were in tight contact with each other, but also globular ones (*ca* 23%) which laid on the flattened ones were observed (Figure 3A,B). The cells showed very sparse microvilli and vesicles on their surface and did not seem to try to invade the Matrigel (Figure 3A,C). However, about 3% of the polygonal cells showed a conic/funnel shape suggesting they were adapting their phenotype to pass through the Millipore filter pores (Figure 3B). Interestingly, LoVo-R cells cultivated on the same substrate exhibited a unique phenotype, appearing as grouped-like regular cobblestone-shaped cells (10–20 µm). They showed a very flattened upper cytoplasmic surface and no tight cell-cell contacts. All cells developed densely and uniformly distributed microvilli and microvesicles on the cytoplasmic surface (Figure 3D–F). The above data suggest that Matrigel mimicking the BM significantly alters the morphological features of both cell types as compared to the same cells cultured on Millipore.

To mimic a thick BM, the LoVo-S/-R cells were cultivated on a Millipore filter covered by a more concentrated Matrigel (3.5 mg/mL). LoVo-S cells showed thick polygonal phenotypes adhering to the Matrigel substrate, which appeared continuous and completely filled the Millipore pores. Notably, they exhibited a significantly higher number of globular cells (34%, *p* < 0.05%) as compared to the same cells cultivated on a standard concentration of Matrigel. Conic/shaped cells (4%) were also detectable (Figure 4A). All the cells showed tight contacts and microvilli on the cytoplasmic surface (Figure 4B,C). LoVo-R cells cultivated on the same substrate included grouped polygonal-shaped cells but also globular ones (60–65%, *p* < 0.05%), which showed only a few tight cell-cell contacts and many microvilli and vesicles on their surface (Figure 4D–F). It is worth noticing that we observed cells exhibiting intercellular tunneling nanotubes with exosomes and microvesicles on their cytoplasmic surface (Figure 4F).

### 3.3. Ultrastructural Morphological Features of LoVo-S and LoVo-R Cells Cultured on Millipore/Collagen Mimicking a Normal and a Desmoplastic Lamina Propria

We further evaluated the ultrastructural cell morphology of LoVo cells cultured on collagen mimicking the normal lamina propria. For this purpose, both LoVo-S and LoVo-R cells were cultivated on a Millipore filter covered with type I collagen at a 0.2 mg/mL concentration. Most LoVo-S cells appeared in tight contacts and grew one on each other. Equally distributed polygonal but also elongated, flattened, or globular-shaped cells were detectable (Figure 5A–C). These cells showed microvilli and sparse microvesicles (Figure 5B,C). Most LoVo-R cells showed a regular polygonal cobblestone shape and appeared next to each other but with no direct cell-cell contacts. About 6% of them appeared only partially flattened and tried to invade the collagen network by assuming a concave cobblestone shape (Figure 5D,E). All LoVo-R cells exhibited many microvilli and microvesicles (Figure 5E,F). The obtained results indicate that the presence of type I collagen as an ECM substrate significantly alters the ultrastructural morphology of both cell types as compared to the cells cultured on Millipore.

To mimic a desmoplastic lamina propria, cells were cultivated on a Millipore filter covered by type I collagen at the highest concentration (3.5 mg/mL). Under these conditions, LoVo-S cells included a layer of polygonal cells in tight cell-cell contact with each other and adhering to the collagen fibrils, which were easily distinguishable. On these cells, many globular ones also grew on the flattened ones (Figure 6A,B). All cells showed sparse microvilli and sporadic microvesicles which were also detectable inside the collagen network, therefore exhibiting a different profile as compared to the same cells in the low-concentrated collagen (Figure 6C). LoVo-R cells included globular-shaped cells next to each other but with few tight cell-cell contacts, so that collagen fibrils were always visible (Figure 6D). All LoVo-R cells, completely covered by microvesicles and microvilli, appeared strongly adherent to the fibrils with filopodia penetrating the collagen fibril network and showed a different pattern from those cultured on the low concentrated collagen (Figure 6E,F).

## 4. Discussion

All cells embedded into the peri-tumoral fibrillar collagen meshwork interplay with ECM through surface receptors like integrins and discoidin domain receptors and respond by synthesizing and secreting different matrix macromolecules [48]. Overproduction of collagens, increased crosslinking, and alignment of collagen fibers induce an increased peri-tumoral stiffness which favors cancer cell invasion and tumor malignancy [29,36,49,50]. Type I collagen new deposition is frequently related to tumor progression, but it has been considered as one of the ECM components also favoring chemoresistance [31,37,38,51]. The possible relation between collagen and drug resistance in CRC cells prompted us to evaluate the behavior and fine morphological characteristics of two cell types: LoVo-S cells, sensitive to doxorubicin, and LoVo-R, resistant to the same anticancer drug. Epithelial cells, like LoVo cells, grow and develop on a very smooth substrate like the BM, which represents the first biological ECM barrier opposing cancer invasion. Any functional interaction between cells and the collagen meshwork within the TME is particularly remarkable when cancer cells degrade and cross the BM. Therefore, to better mimic the in vivo conditions, we cultivated LoVo cells in a three-dimensional culture of Millipore filter coated by different concentrations of Matrigel, mimicking both normal and thicker BM or by two different concentrations of collagen to mimic the deeper barrier of a normal lamina propria (low concentration) or a desmoplastic one (higher concentration), which develops in colon wall as the tumor evolves and grows. When LoVo cells perforate the smooth and planar BM and invade a desmoplastic lamina propria, they find a new unknown microenvironment, where the rough meshwork of fibrils composing collagen fibers constitutes the major component of ECM. LoVo cells cultivated only on a Millipore filter with 5 µm pore size, which does not allow the cells to pass through, showed different phenotypes and behavior at SEM: LoVo-R cells, developing filopodia and assuming a migrating concave conic or funnel shape, exhibited an evident trend to invade and cross the filter pores when compared to LoVo-S cells. Similarly, real-time qPCR analysis of LoVo-S and LoVo-R cultivated on a Millipore filter covered by different concentrations of Matrigel or type I collagen demonstrated that LoVo-S cells are basically less aggressive than LoVo-R cells considering E-cadherin, vimentin, and Snail [52].

The LoVo cancer cell invasion of ECM completely takes place when the same cells invade the collagen network of a desmoplastic lamina propria. In this experimental setup, LoVo-S cells grown on the highly concentrated type I collagen substrate (mimicking a desmoplastic lamina propria) showed a strong increase of E-cadherin expression, thus apparently suggesting that a high collagen network can reduce the LoVo-S aggressiveness. However, the same substrate seems to increase LoVo-S aggressiveness as it stimulated Snail expression as well as a strong increase of the expression of MMP-2, -9, -14, and HPSE. Clinical studies report that tumor growth and metastasis can develop with or also without E-cadherin deregulation [53], and a loss of the E-cadherin expression is not necessary to evoke distant metastasis [54,55,56,57]. Due to the plasticity of invasion, it was reported that also the epithelial MCF-7 breast cancer cells, which do not show an evident MMP-14 expression, are indeed able to invade a lower dense collagen network collectively. Therefore, it has been recently suggested that ECM confinement, more than E-cadherin downregulation, can mechanically control ECM invasion [58]. In a previous study it was reported that cancer cells of solid tumors could lose their epithelial phenotype by developing only “partial EMT” phenotypes which migrate in group as clusters in vivo [59]. Notably, it has also been supposed that an epithelial/mesenchymal plasticity in metastatic dissemination of cancer cells might not be driven only by a full EMT but also by a hybrid epithelial/mesenchymal state of cancer cells. In this case, cells co-express both epithelial and mesenchymal markers or even develop a mesenchymal-to-epithelial transition (MET) phenotype, which might contribute to initiating different tumors and favoring metastasis [60].

Considering the influence of the substrate type on LoVo-S cells properties, highly concentrated collagen, corresponding to a desmoplastic lamina propria in CRC, strongly induces an increase of EMT markers. However, both the MMP-9 and HPSE, whose expression is upregulated in many primary human tumors, including CRC, and plays a primary role in tumor development and metastasis [61,62], were moderately expressed also in concentrated Matrigel samples. This suggests that in LoVo-S cells, both MMP-9 and HPSE expressions could depend much more on the thickness/concentration of the substrate as compared to the other matrix effectors described above. The ultrastructure of LoVo-S/-R cells cultivated on a plastic dish in two-dimensional cultures showed different phenotypes, which were partially confirmed by our three-dimensional cultures on the Millipore filter alone [63]. The morphological phenotype of LoVo-S cells cultivated on different substrates confirmed the data of the real-time qPCR analysis: polygonal, flattened, elongated, or globular-shaped cells were observed in different percentages, with the globular phenotype much more present in highly concentrated collagen samples. However, the LoVo-S cells always exhibited cell-cell contacts and did not clearly invade the ECM substrates even when they developed an elongated mesenchymal phenotype like in low concentrated collagen groups or a globular shape secreting exosomes and microvesicles in the highly concentrated collagen meshwork.

Differently, when LoVo-R cells started to invade ECM components, neither the substrate type nor the concentration seemed to significantly affect E-cadherin, vimentin, Snail, or MMP-2, -14 expressions which exhibited similar values in all samples. HPSE expression showed only a very little increase in highly concentrated collagen, whereas the highest MMP-9 expression measured in highly concentrated collagen samples was also high in concentrated Matrigel groups. Similar to LoVo-S cells, MMP-9 seems to be unaffected by the type of ECM component in LoVo-R cells but depends on the thickness/concentration of the culture substrate. Ultrastructural observations of LoVo-R cells cultivated on a Millipore filter alone showed a concave cobblestone or also a funnel-shaped phenotype with no cell-cell contacts and single filopodia which partially penetrated the filter pores. The cobblestone phenotype was constantly observed when the LoVo-R cells were cultivated on standard Matrigel and low-concentrated collagen. Only when they were in contact with thick Matrigel and highly concentrated collagen substrate, they developed a globular shape with no cell-cell contacts. In highly concentrated collagen, the predominating globular phenotype also developed filopodia which still only partially invaginated the micropores of the collagen fibril meshwork. Thus LoVo-R cells cultivated on different substrates seem not to change their intrinsic original invading capability. The globular phenotype in LoVo-R cells, which in melanoma was considered a potential ameboid migrating one also secreting high level of MMP-9 [64,65], could be related to the higher expression of MMP-9 measured in thicker Matrigel and highly concentrated collagen. As a matter of fact, a similar globular phenotype was also observed in LoVo-S cells growing in contact with the highly concentrated collagen (3.5 mg/mL), which compared to the low concentrated one (0.2 mg/mL), showed larger micropores and thicker fibrils (to distinguish the different aspects of low and high concentrated collagen meshworks compare the Figure 5D,E with Figure 6D,E). It could be plausible that also the larger pores and thicker fibrils of the highly concentrated collagen meshwork contribute to the development of globular LoVo-S cells shedding exosomes and microvesicles in the microenvironment or globular LoVo-R cells with invaginating filopodia.

The first interesting event described next to the tumor mass includes a desmoplastic deposition of collagen, which might correspond to a physical defense of the host [66]. However, in human CRC, the highly expressed type I collagen and collagen fiber alignment also act as promoters of aggressiveness by inducing EMT, tumor progression, and poor patient disease-free survival [25,32,37,51,67]. Notably, the dense and stiff collagen type I fibers stimulate pro-tumorigenic signaling cascades in cancer, such as focal adhesion kinase (FAK), Src kinases family (SFKs), and extracellular regulated kinase (ERK)1/2 [67]. It has also been reported that the interplay of cancer cells with stroma cells and changes in ECM components might contribute to the development of a drug resistance mechanism [68], as demonstrated in MDA-MB-231 breast cancer cells which matured a stiffness-dependent resistance to doxorubicin [69]. The fact that LoVo-R cells’ behavior does not depend in general on both substrate type and concentration and cells show similar values in the different groups might also be related to the intrinsic resistance to doxorubicin which has been reported to be also associated with peri-tumoral collagen deposition [69].

It is plausible to suggest that LoVo-S/-R cells could sense a thicker, stiffer, and larger microporous TME, which could induce them to invade ECM by increasing MMPs and HPSE expressions. It is noteworthy that HPSE, a well-known component of exosomes/microvesicles, can favor both the production and docking of exosomes, thus playing a critical role in both tumor progression and chemoresistance [61]. Strong interactions between chemoresistance and peri-tumoral ECM composition or physical array alterations have been recently suggested: remodeling of ECM and prolonged chemotherapy can favor treatment resistance and ECM-cell communications [31,37,38,51,70,71,72,73]. In this study, we demonstrated that the CRC LoVo-S cells are affected by highly concentrated collagen, but once they become doxorubicin-resistant (LoVo-R cells), they maintain their intrinsic aggressiveness and are independent of the type of ECM substrate (Matrigel or collagen). The data are summarized in Figure 7, where the different effects of the concentrated collagen substrates on gene expression and morphological phenotype in LoVo-S/-R cells are shown.

In summary, the highly concentrated collagen meshwork mimicking the desmoplastic lamina propria in CRC induces LoVo-S cells to develop a hybrid EMT phenotype. On the contrary, LoVo-R cells are not significantly affected by all the tested ECM components: only an increase of the Matrigel substrate thickness or the thicker fibrils and larger micropores in highly concentrated collagen meshwork slightly stimulate the MMP-9 expression and could induce the development of a potential ameboid migrating globular phenotype.

Among the limits of this paper, we remember that a Western blot analysis was carried out only for E-cadherin, as it was not easy to collect the necessary amount of protein material to evaluate the secretion of MMPs from cells adhering and growing on matrix substrates. Moreover, it is probable that the gene expression and complete secretion of the same proteins could need different experimental times. We think that future studies, considering the different longer time of culturing, could help us in collecting other EMT markers at the protein level to confirm our current molecular data.

Moreover, even though LoVo-S/-R cells are commonly considered two cell lines for their different aggressiveness and leading anticancer therapies are individually adapted to the single patient characteristics, the reported data should also be confirmed in other cell lines. Therefore, to better understand whether the proposed model could also be applied to other colon cancer cell types, we aim to relate our data with future studies on primary cancer cells taken from patients and cultivated on wider pore filters which notably better allow a free cancer invasion of the ECM.

## Figures and Tables

**Figure 1 biomolecules-12-01786-f001:**
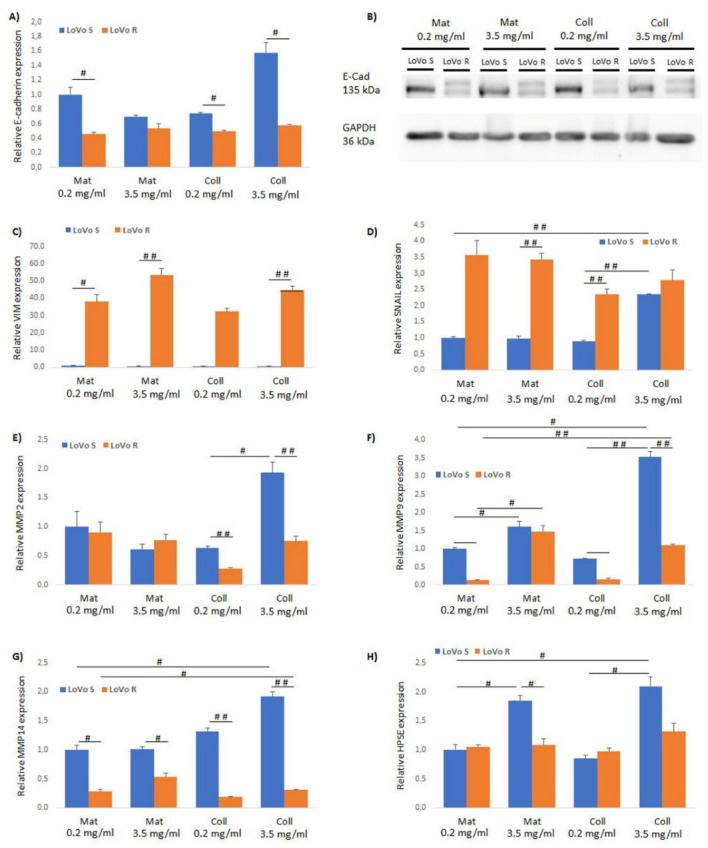
Evaluation of EMT markers and matrix effectors in LoVo-S and LoVo-R cells. Cells were cultured for 24 h on a Millipore filter with 5 µm pores in the presence of matrix substrates (Matrigel and collagen) at different concentrations (0.2 and 3.5 mg/mL). In LoVo-S cells, E-cadherin (**A**,**B** using RT-PCR and western blot, respectively), Snail, MMP-2, -9, and -14 gene expressions were highly increased in concentrated collagen substrate (3.5 mg/mL) (**D**–**G**). Both in Matrigel and type I collagen cultures of LoVo-S cells, the HPSE expression depends on substrate concentration and is independent of substrate type (**H**). In LoVo-R cells, E-cadherin, vimentin, Snail, MMP-2, and -14 expressions are independent of both the type and concentration of the substrate (**A**–**E**,**G**,**H**). Both in Matrigel and type I collagen culture LoVo-R cells, MMP-9 expression depends on substrate concentration and is independent of substrate type (**F**). Bars represent the mean ± standard deviation (SD). The gene expression was normalized to GAPDH as a housekeeping gene. **^#^**
*p* ≤ 0.05; **^##^**
*p* ≤ 0.001. Statistical analyses were performed using the Relative Expression Software Tool (REST).

**Figure 2 biomolecules-12-01786-f002:**
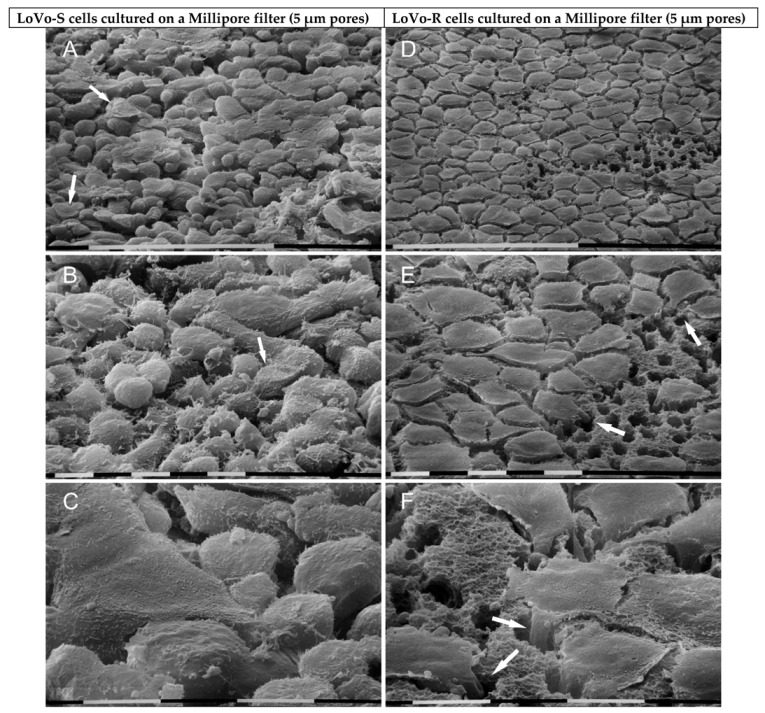
Ultrastructural morphology of LoVo-S and LoVo-R cells cultured for 24 h on a Millipore filter. LoVo-S cells show grouped flattened cells but also globular ones. All of them display cell-cell contacts. A number of cells (6%) exhibit a concave conic/funnel shape (arrows) when trying to pass through the Millipore pores (**A**,**B**). All cells show few microvilli and no microvesicles on their surface (**C**). LoVo-R cells look like smooth, regular, and very flattened cobblestone-shaped cells. They appear next to each other but do not show tight cell-cell contacts. A tilted observation demonstrates that they appear like concave leaves and develop filopodia (arrows) crossing the pores of the Millipore filter (**D**–**F**). Bar = 100 µm (**A**,**D**). Bar = 10 µm (**B**,**C**,**E**,**F**).

**Figure 3 biomolecules-12-01786-f003:**
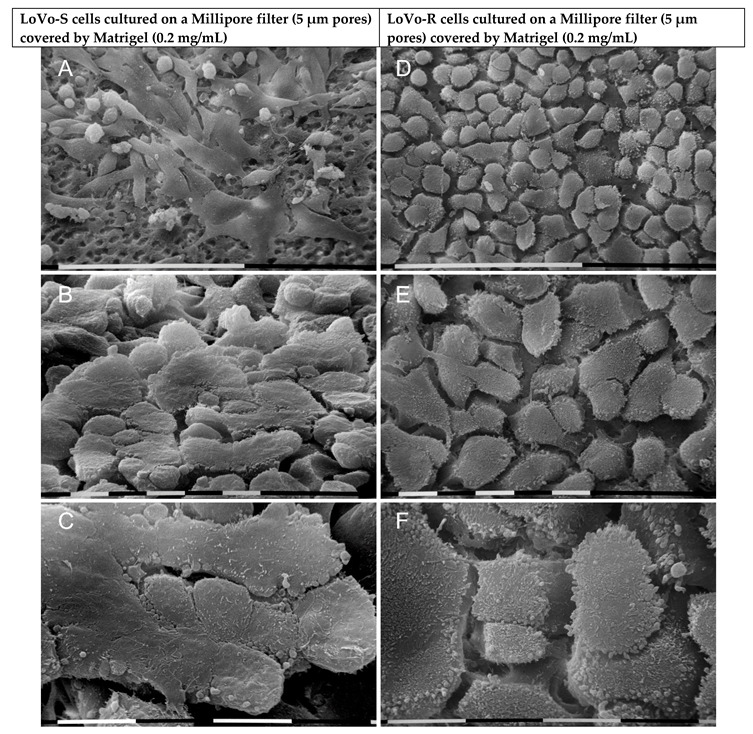
LoVo-S and LoVo-R were cultivated for 24 h on a Millipore filter with 5 µm pores covered by Matrigel at standard concentration mimicking the BM. LoVo-S cells show large and very flattened polygonal cells (10–20 µm) in tight contact with each other, but also a few globular ones (10 µm in diameter), which grow on the flattened ones (**A**). Sparse microvilli and a few vesicles are observable on their surface (**B**,**C**). Only a few (*ca* 3%) of the polygonal cells developed a conic/funnel shape, which suggests they were trying to invade the Matrigel (**B**). All LoVo-R cells appear like grouped cobblestone-shaped cells (10–20 µm) with a very flattened upper cytoplasmic surface and no tight contacts as an intercellular space wide 2–5 µm are always visible. Densely distributed microvilli and microvesicles are evident on their cytoplasmic surface. (**D**–**F**). Bar = 100 µm (**A**,**D**) Bar = 10 µm (**B**,**C**,**E**,**F**).

**Figure 4 biomolecules-12-01786-f004:**
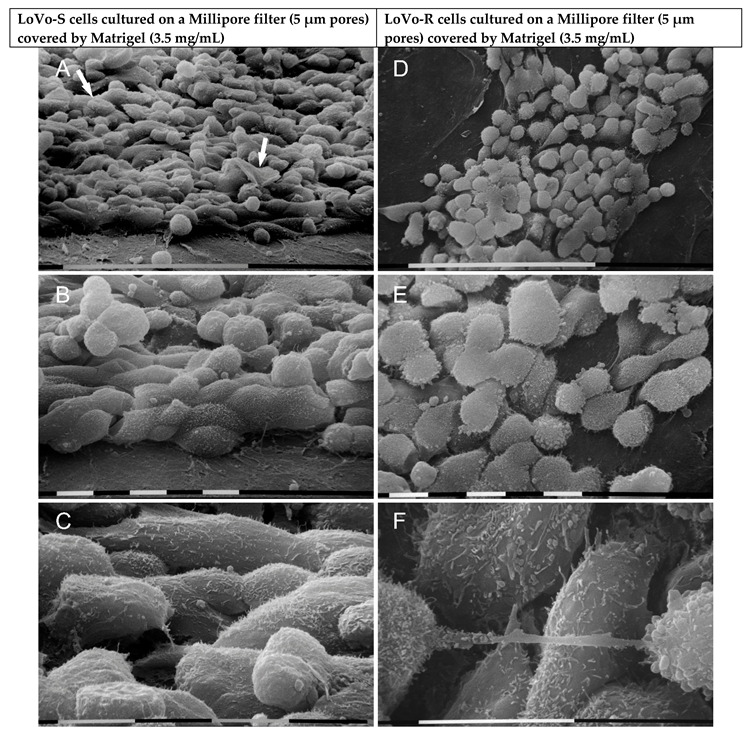
Ultrastructural morphological features of LoVo-S and LoVo-R cells cultivated for 24 h on Millipore/concentrated Matrigel mimicking a thick basement membrane. LoVo-S cells appeared as thick polygonal-like cells, but many globular ones adhering to the substrate were present. Some conic/funnel-shaped cells (*ca* 4%) are visible (arrows) (**A**). Matrigel substrate appears continuous and completely fills the Millipore pores (**A**,**B**). Cells exhibit microvilli on the cytoplasmic surface (**B**,**C**). LoVo-R cells, on the other hand, show both grouped polygonal or globular cells (*ca* 60–65%, *p* < 0.05) with few tight contacts and many microvilli and vesicles on their cytoplasmic surface (**D**–**F**). Matrigel substrate completely fills the filter pores (**D**,**E**). Among two adjacent cells with many microvilli and microvesicles, an intercellular tunneling nanotube with exosomes and microvesicles on its cytoplasmic surface is visible (**F**). Bar = 100 µm (**A**,**D**). Bar = 10 µm (**B**,**C**,**E**,**F**).

**Figure 5 biomolecules-12-01786-f005:**
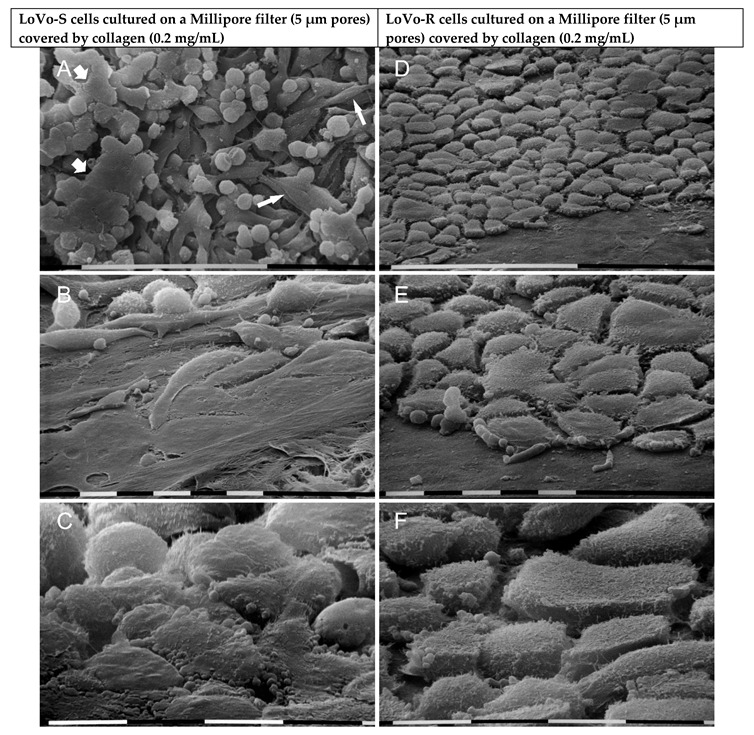
Morphology of LoVo-S and LoVo-R cells cultured for 24 h on Millipore/type I collagen mimicking the normal lamina propria. Grouped LoVo-S cells include polygonal and some elongated cells (narrow arrows) but also globular ones growing on the firsts and appearing in tight contact one to each other so as to seem like to be fused together (large arrows) (**A**–**C**). The cells exhibit microvilli but few microvesicles (**B**,**C**). LoVo-R cells look like polygonal and partially flattened shaped cells with no tight cell-cell contact and prepare themselves to invade the collagen substrate by developing a concave cobblestone shape (**D**,**E**). All the cells clearly include microvilli (**E**,**F**). Bar = 100 µm (**A**,**D**). Bar = 10 µm (**B**,**C**,**E**,**F**).

**Figure 6 biomolecules-12-01786-f006:**
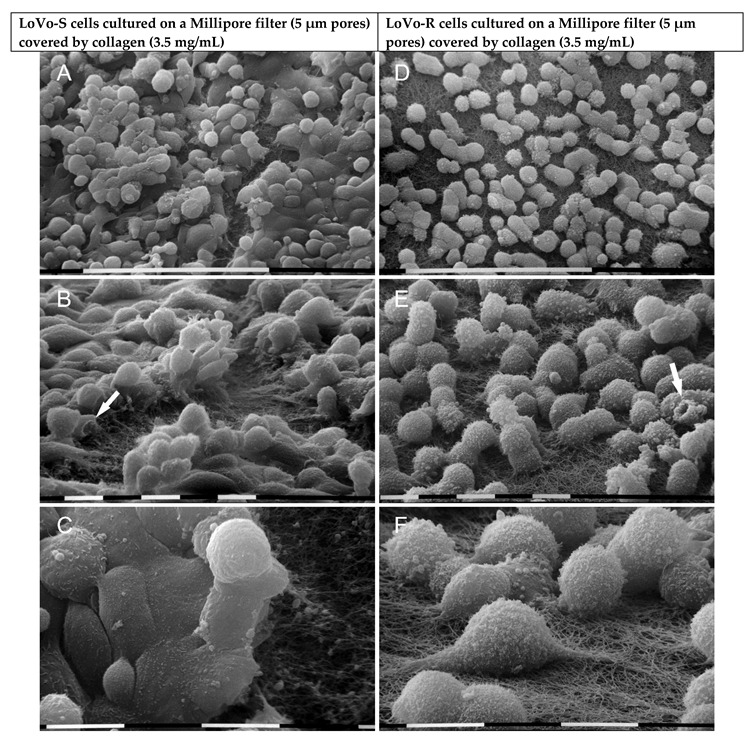
Ultrastructural morphological characteristics of LoVo-S and LoVo-R cells cultured for 24 h on Millipore/type I collagen mimicking a desmoplastic lamina propria. LoVo-S cells include polygonal cells in tight cell-cell contact with each other and adhering to the collagen fibrils, which are only partially visible. However, globular-shaped cells grow on the flattened ones (**A**,**B**). A funnel-shaped cell is detectable (arrow) (**B**). LoVo-S cells show few microvilli, but exosomes and microvesicles are embedded in the collagen fibril network (**C**). The LoVo-R cells look like globular-shaped cells next to each other, but no tight contacts are visible, so collagen fibrils are always distinguishable (**D**). A funnel-shaped cell is detectable (arrow) (**E**). The LoVo-R cells develop many microvesicles on their surface and appear strongly adherent to the fibrils with also filopodia invaginating the micropores of the collagen fibril network (**E**,**F**). Bar = 100 µm (**A**,**D**) Bar = 10 µm (**B**,**C**,**E**,**F**).

**Figure 7 biomolecules-12-01786-f007:**
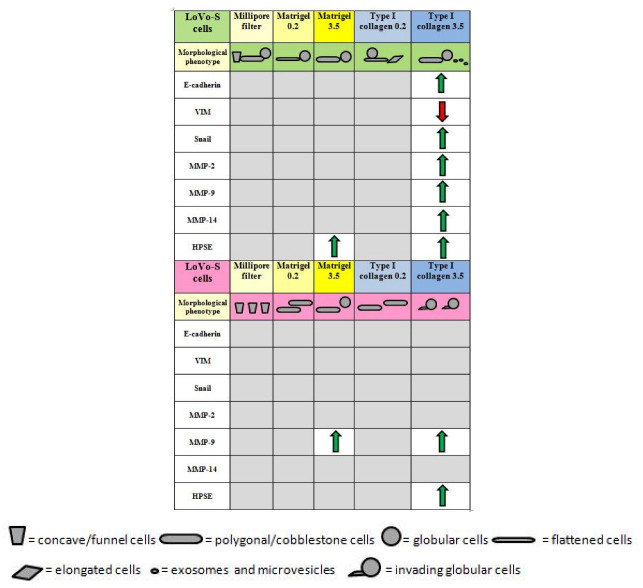
Effects of different substrates on gene expression and morphological phenotype in Lovo-S/-R cells. The green arrows show an increase in the EMT markers or matrix-degrading enzymes, whereas the red one means a decrease in the mesenchymal marker vimentin. Grey zones represent no-significant differences among the different substrates.

**Table 1 biomolecules-12-01786-t001:** List of real-time qPCR primers used in this study.

Gene	Primer Sequence
E-Cadherin	F: TTCTGCTGCTCTTGCTGTTT, R: TGGCTCAAGTCAAAGTCCTG;
Vimentin (VIM)	F: AAAACACCCTGCAATCTTTCAGA, R: CACTTTGCGTTCAAGGTCAAGAC;
SNAIL	F: AGTTTACCTTCCAGCAGCCCTAC, R: AGCCTTTCCCACTGTCCTCATC;
MMP-2	F: TGCATCCAGACTTCCTCAGGC, R: TCCTGGCAATCCCTTTGTATGTT;
MMP-9	F: GGTGATTGACGACGCCTTTG, R-CTGTACACGCGAGTGAAGGT;
MMP-14	F: TGCCATGCAGAAGTTTTACGG, R: TCCTTCGAACATTGGCCTTG;
Heparanase (HPSE)	F: ATTTGAATGGACGGACTGC R: GTTTCTCCTAACCAGACCTTC;
GAPDH	F: ACACCCACTCCTCCACCTTT R: TCCACCACCCTGTTGCTGTA;

## Data Availability

The original contributions presented in the study are included in the article material. Further inquiries can be directed to the corresponding author.

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
