# Peer review of "Substrate Type and Concentration Differently Affect Colon Cancer Cells Ultrastructural Morphology, EMT Markers, and Matrix Degrading Enzymes"

_biomolecules, 2022, doi:10.3390/biom12121786_

Round 1
Reviewer 1 Report
In the present work, the authors evaluate the effect of different substrates type and concentrations on ultrastructural morphology of colon cancer cell line LoVo that are or not resistant to doxorubicin. Matrigel and type I collagen have been evaluated in a model that mimics the structural ECM barriers. This model is based on the growth of cells on a commercial filter with a pore size of 5 µm. The filter is coated with Matrigel or type I collagen at two concentrations and, in addition to determining the protein levels of E-cadherin and expression levels of E-cadherin and other genes of EMT markers and implicated in metastasis, morphological changes at the structural level are also evaluated by scanning electron microscopy. LoVo-R is used as a model of colon cancer cells near the metastatic state.
LoVo-S cells are more affected by collagen at high concentration, developing a hybrid EMT phenotype, while LoVo-R shows very few changes in the evaluated ECM components.
Manuscript is well writing but some misspellings must be corrected.
The manuscript shows interesting data but the research objective is not well formulated or justified. It may be that the model used, LoVo cells, a line coming from a metastatic tumor, is not the most appropriate for the objective to be evaluated. It would be better to use a line coming from a primary tumor and to confront it with the same line resistant to treatment and possibly closer to having metastatic characteristics.
Major points:
The main problem of the manuscript is that the model evaluated needs to be more justified: why these filters have been used? Why 5 µm pore size? Was the model used in other type of cells? Are there other alternatives to this model? Why LoVo cells are used?
Minor points:
In the western blot and RT-qPCR cells without treatment haven’t been evaluated despite that these groups are included in the electron micrography.
Statistically analysis section must be included.
Please correct the text.
Abbreviations must be used the first time that the acronym appears.
Author Response
Review 1 Report Form
Open Review
Comments and Suggestions for Authors
In the present work, the authors evaluate the effect of different substrates type and concentrations on ultrastructural morphology of colon cancer cell line LoVo that are or not resistant to doxorubicin. Matrigel and type I collagen have been evaluated in a model that mimics the structural ECM barriers. This model is based on the growth of cells on a commercial filter with a pore size of 5 µm. The filter is coated with Matrigel or type I collagen at two concentrations and, in addition to determining the protein levels of E-cadherin and expression levels of E-cadherin and other genes of EMT markers and implicated in metastasis, morphological changes at the structural level are also evaluated by scanning electron microscopy. LoVo-R is used as a model of colon cancer cells near the metastatic state.
LoVo-S cells are more affected by collagen at high concentration, developing a hybrid EMT phenotype, while LoVo-R shows very few changes in the evaluated ECM components.
Manuscript is well writing but some misspellings must be corrected.
The manuscript shows interesting data but the research objective is not well formulated or justified. It may be that the model used, LoVo cells, a line coming from a metastatic tumor, is not the most appropriate for the objective to be evaluated. It would be better to use a line coming from a primary tumor and to confront it with the same line resistant to treatment and possibly closer to having metastatic characteristics.
We thank the reviewer for finding our research work interesting and the constructive comments made. The major point of this experimental study was to evaluate the cell morphology using SEM and to understand at cellular level the importance of dox resistance in cell phenotype in colon cancer, a disease with a high mortality rate, taking also into account the chemo-resistant to dox. Below please find our responses to the comments of the reviewer. Once more thanks for the comments which helped us to improve the quality of the manuscript.
Regarding the cell lines point we certainly agree for a primary cell specimen to be examined and it may be of significant value to be considered in future studies in collaboration with clinicians, an issue we plan for. Here we would like to point out that the different cancer cell lines are often difficulty comparable in their behavior. Moreover, even the same cancer cell line can express different responses when analyzed in the same experimental conditions thanks to the plasticity property. Another crucial point is that the current anticancer therapies are individually adapted to the single patient characteristics. Looking for a common behavior among different cancer cell lines could make more difficult the result interpretations at the present time. These were the main reason we used the well-established model of LoVo sensitive and resistant to Dox cells. We have accordingly added in discussion the sentence: “Following these critical initial evaluations, in respect to the role of ECM substrates in LoVo-S/-R cells, future studies using primary specimens from patients as well as other cell lines will help us to further evaluate the importance of ECM substrates in EMT at cellular and molecular level”.
Major points:
The main problem of the manuscript is that the model evaluated needs to be more justified: why these filters have been used? Why 5 µm pore size?
Indeed, this is a very nice point, thanks you. As it is stated in the manuscript, we wanted to observe the behavior of cancer cells preparing themselves to invade different ECM substrates, at different concentrations. Millipore filters with 5 µm pores is a selective physical barrier able to do not allow cancer cells to migrate through the pores; any way cells are free to differently express EMT markers. Using traditional Millipore filters with 8 µm pore we lost many cells which passed through the filter after 24 hours. This has been justified in more detail in the revised manuscript in the end of Introduction and Materials and Methods (Cell Cultures) sections Discussion, accordingly.
Was the model used in other type of cells? Are there other alternatives to this model?
The 5 µm pore filters are often used in 3D cultures to avoid cells migration by keeping all the cells on the filter. In our model the cell can freely change the expression of EMT markers as they are in contact with the different ECM substrates and also can feel the input to migrate/invade. This has now been added in the revised manuscript discussion part, accordingly.
Why LoVo cells are used?
Among the many available cell lines, we chose the LoVo cell line derived from colon cancer because it represents a widely used experimental model for the study of EMT (more than 80 articles on Pubmed for "LoVo cells and EMT"). In addition, LoVo cells and its multidrug-resistant LoVo-R subline, obtained by prolonged exposure to doxorubicin, are often used for the study of drug resistance in cancer (more than 140 articles on Pubmed for "LoVo cells and multidrug resistance"). Considering that most patients with CRC develop resistance to chemotherapeutic drugs, this experimental model seemed useful to us for a more comprehensive study of EMT in colon cancer cells.
Minor points:
In the western blot and RT-qPCR cells without treatment haven’t been evaluated despite that these groups are included in the electron micrography.
We thank for this comment too. We wanted to evaluate the different behavior of LoVo cells in different biological substrates to better mimic the in vivo conditions. The only Millipore substrate was used as morphological control just to underline that the physical aspect, beside the chemical composition, can affect the cancer cell activity. The control group for the western blot and RT-qPCR could be represented by cell growing on a normal basement membrane (Matrigel 0.18 mg/ml).
Statistically analysis section must be included.
Thank you for remembering us. We added a short Section.
Please correct the text.
Abbreviations must be used the first time that the acronym appears.
Yes it is right. We corrected.
Reviewer 2 Report
The authors use the LoVo CRC cancer model to study the EMT phenotype associated with resistance to doxorubicin and their resulting phenotype when grown on different 'matrix like' in vitro substrates such as matrigel and type 1 collagen.
Overall the visual differences in the EM images are striking and look really beautiful. They also find a large difference in EMT phenotype using qPCR.
My two main concerns are:
1) Western-blot was only performed for e-cad. I think it should be performed for all of the associated genes investigated (vim, snail, mmp2, 9 and 14). qPCR results while significant can sometimes not translate to the protein level and thus it is key that this experiment is performed.
2) . The use of only one cell line. I understand the scope of this study was to use EM to visualize the difference in sensitive vs resistant cells and it requires a lot of imaging with one cell line but I believe a minimum of two cell lines should have been used for this study to at least get a sense of changes that are consistent vs different across multiple lines given the heterogeniety of patient CRC in response to therapy.
Small minor changes include:
1) In the introduction, please include the common chemotherapies patients often receive (what is frontline), statistics of patients who become resistance and median time to resistance if possible.
2) Does an EMT phenotype change proliferation in CRC cells and do resistance cells often show a slower-cycling phenotype as seen in other cancers? One sentence in the intro will suffice
3) The authors say collagen 1 deposition physically protects cells. Can they elaborate on this, how does it physically protect the cells?
4) the E-cad westernblot has writing cut off on the side.
Author Response
Review 2 Report Form
Comments and Suggestions for Authors
The authors use the LoVo CRC cancer model to study the EMT phenotype associated with resistance to doxorubicin and their resulting phenotype when grown on different 'matrix like' in vitro substrates such as matrigel and type 1 collagen.
Overall the visual differences in the EM images are striking and look really beautiful. They also find a large difference in EMT phenotype using qPCR.
We thank the reviewer for the good words and comments and for finding our research work striking. The major issue we face was indeed to evaluate the cell morphology using SEM and to understand at cellular level the importance of dox resistance in cell phenotype in a dramatic disease like colon cancer with a high mortality rate. Below our responses to the comments of the reviewer which helped us to improve the quality of the manuscript, accordingly.
My two main concerns are:
1) Western-blot was only performed for e-cad. I think it should be performed for all of the associated genes investigated (vim, snail, mmp2, 9 and 14). qPCR results while significant can sometimes not translate to the protein level and thus it is key that this experiment is performed.
We certainly agree with the reviewer regarding the EMT markers. Indeed, the alteration in e-Cadherin at both mRNA and protein levels, as a crucial point to evaluate EMT, prompted us to the current proposal. All other markers that have been evaluated at mRNA level exhibited a profile that follow what was expected. However, due to significantly low amount we are able to isolate it is not easy to proceed further with the evaluation at protein level using western blotting. This is especially very difficult when cells grown and adhere on matrix substrates. We added a sentence in the discussion part to declare this point: “Evaluation of other EMT markers at protein level as to further elucidate the data at protein level will be useful to clearly demonstrate the data at molecular level. Future studies and in correlation with other cell lines will further help to understand this issue and whether the proposed model is also applied in other colon cancer cell types”.
2) The use of only one cell line. I understand the scope of this study was to use EM to visualize the difference in sensitive vs resistant cells and it requires a lot of imaging with one cell line but I believe a minimum of two cell lines should have been used for this study to at least get a sense of changes that are consistent vs different across multiple lines given the heterogeneity of patient CRC in response to therapy.
We thank the reviewer also for this important comment. We had the same thoughts. However, at this stage we decided to present these promising results with such striking findings. Presenting other cell lines, that we are planning for future research purposes in a continuation of this project, will result to many pictures and experimental groups. This will make more difficult to understand the differences at this stage before presenting the main finding and may confuse the reader and the clear current message we present in the manuscript. This also requires a lot of work and time. Indeed, here we answer that the same cancer cell line can show different behavior due to plasticity and moreover when they grow in different ECM substrates. Therefore, we focused our aim on LoVo cells because they are considered two cell lines for their very different aggressiveness. Authors in line with this reviewer will work in this issue for future studies. This point is now included in the discussion as given above in point 1.
Small minor changes include:
Many thanks also for the minor points below. We have incorporated the revisions in the respective places in the revised manuscript.
1) In the introduction, please include the common chemotherapies patients often receive (what is frontline), statistics of patients who become resistance and median time to resistance if possible.
In the Introduction Section we added some sentences and two new references about the CRC:
“The final rate survival depends on the stage of colon cancer: the 5-year survival rates of stage I–IV colon cancer are 96.6%, 88.7%, 69.9%, and 34.3%, respectively. Anyway, the adjuvant chemotherapy after radical surgery seems to improve the survival rate of stage II treated patients (90.4%) vs. stage II untreated ones (82.4%) (5). However, during the chemotherapy the colon cancer patients can develop a drug resistance which can lead to the failure of treatment (6-7). For instance, more than 50% of metastatic colorectal cancer patients develop a resistance to 5-fluorouracil, one of the most effective and most commonly selected drugs to treat colorectal cancer (8).”
2) Does an EMT phenotype change proliferation in CRC cells and do resistance cells often show a slower-cycling phenotype as seen in other cancers? One sentence in the intro will suffice
We thank the reviewer for this point of reflection. In the Introduction Section we completed a sentence by adding what was suggested in Reviewer’s comments.
3) The authors say collagen 1 deposition physically protects cells. Can they elaborate on this, how does it physically protect the cells?
Type I collagen at first opposing cancer cell invasion is considered as a physical natural barrier. This is because it is often associated to stiff and large diameter fibrils which then array in densely packed circular fibers surrounding tumor. Successfully, fibers change their array and favor cancer cell invasion.
4) the E-cad westernblot has writing cut off on the side.
Thank you. We corrected the figure.
Round 2
Reviewer 1 Report
The manuscript is suitable for publication in the present form.
Author Response
We thank again the Reviewer for helping and guiding us to improve the understanding and quality of this paper.
Reviewer 2 Report
My two major concerns were discussed by the authors:
1) I understand the difficulty of collecting enough protein from proper western analysis in this model system. I feel like you should discuss this shortcoming in the disucssion.
2) I agree that direct comparisons between cell lines are tough and are planned for a future paper. Again, I feel a discussion paragraph is needed explaining some of the limitations of the study and how you plan to address them properly in future manuscripts.
If these are done appropriately along with addressing the other reviewers and editors concerns, this paper is acceptable for publication.
Author Response
My two major concerns were discussed by the authors:
1) I understand the difficulty of collecting enough protein from proper western analysis in this model system. I feel like you should discuss this shortcoming in the discussion.
2) I agree that direct comparisons between cell lines are tough and are planned for a future paper. Again, I feel a discussion paragraph is needed explaining some of the limitations of the study and how you plan to address them properly in future manuscripts.
Responses to Reviewer 2 Comments:
We thank again the Reviewer for helping us in improving the quality of the paper. Therefore, following the Reviewer requests we added some paragraphs at the end of the Discussion section:
“Among the limits of this paper we remember that a Western-blot analysis was carried out only for E-cadherin, as it was not easy to collect the necessary amount of protein material to evaluate the secretion of MMPs from cells adhering and growing on matrix substrates. Moreover, it is probable that the gene expression and complete secretion of the same proteins could need different experimental times. We think that future studies, considering different longer time of culturing, could help us in collecting other EMT markers at protein level to confirm our current molecular data.
Moreover, even though LoVo-S/-R cells are commonly considered two cell lines for their different aggressiveness and leading anticancer therapies are individually adapted to the single patient characteristics, the reported data should be confirmed also in other cell lines. Therefore, to better understand whether the proposed model could be also applied in other colon cancer cell types, we aim to relate our data with future studies on primary cancer cells taken from patients and cultivated on wider pore filters which notably better allow a free cancer invasion of the ECM.”
If these are done appropriately along with addressing the other reviewers and editors concerns, this paper is acceptable for publication.